# Combating Noise: Semi-supervised Learning by Region Uncertainty Quantification

**Zhenyu Wang**     **Yali Li** *     **Ye Guo**     **Shengjin Wang**

Beijing National Research Center for Information Science and Technology (BNRist)
Department of Electronic Engineering, Tsinghua University
{wangzy20, guo-y18}@mails.tsinghua.edu.cn
{liyali13, wgsgj}@tsinghua.edu.cn

## Abstract

Semi-supervised learning aims to leverage a large amount of unlabeled data for performance boosting. Existing works primarily focus on image classification. In this paper, we delve into semi-supervised learning for object detection, where labeled data are more labor-intensive to collect. Current methods are easily distracted by noisy regions generated by pseudo labels. To combat the noisy labeling, we propose noise-resistant semi-supervised learning by quantifying the region uncertainty. We first investigate the adverse effects brought by different forms of noise associated with pseudo labels. Then we propose to quantify the uncertainty of regions by identifying the noise-resistant properties of regions over different strengths. By importing the region uncertainty quantification and promoting multi-peak probability distribution output, we introduce uncertainty into training and further achieve noise-resistant learning. Experiments on both PASCAL VOC and MS COCO demonstrate the extraordinary performance of our method.

## 1 Introduction

Deep neural networks (DNNs) have developed significantly in the computer vision area [15, 18]. Despite this, DNNs highly rely on the fully-supervised learning with a large amount of human-annotated data, which consumes a tremendous amount of time to annotate. In comparison, unlabeled images are much easier to access. Semi-supervised learning [4] is thus studied to address this problem. By involving large-scale unlabeled images in training, semi-supervised learning becomes more valued [16, 26, 40, 28] on benchmark datasets.

However, most of the existing semi-supervised learning methods focus on image classification. Object detection, where complete annotations include category-aware tags and location-aware bounding boxes, requires much more efforts to construct large-scale fully-annotated datasets. In this work, we aim at semi-supervised learning for object detection [35]: an object detector is trained on a dataset where only a small fraction of images are fully-labeled and the rest of them are unlabeled. In this setting, easily obtained unlabeled data are utilized to improve the performance of fully-supervised object detection. Most of the current semi-supervised learning methods in object detection are based on pseudo labeling [19, 36]. A fully-supervised model is firstly pre-trained on completely labeled images then performs inference on unlabeled data to generate pseudo labels. These unlabeled images associated with pseudo labels are further combined with labeled data to learn the semi-supervised model. This pseudo labeling scheme has been widely adopted in semi-supervised object detection [31]. However, the performance is still limited since the noise inherently exists in pseudo labels, i.e., pseudo labeling based semi-supervised learning methods severely suffer from the inherent noise.

---

*Corresponding author

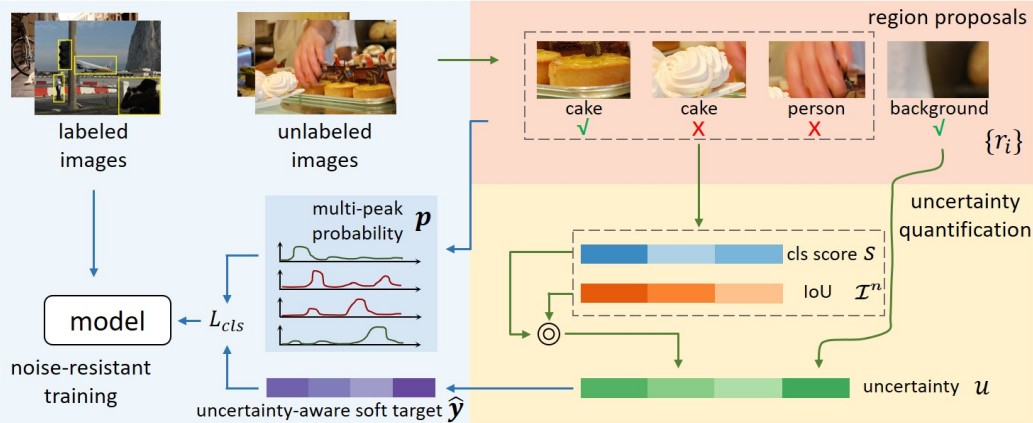

Figure 1: **The working framework of our proposed method.** To combat noise associated with pseudo labels, we quantify uncertainty for different regions first. By further constructing uncertainty-aware soft target and promoting multi-peak probability distribution, we introduce uncertainty into training and achieve noise-resistant learning.

To promote semi-supervised learning, it is natural to ask: *what kinds of adverse effects are caused by the noisy pseudo labels in detection?* A common way of object detection is to firstly generate several candidate region proposals then extract object-centric feature representations. Because of the noisy pseudo labels, region proposals are likely to be assigned with incorrect labels. Based on the error analysis, we discover that three kinds of errors (i.e., the missed GT error, the classification error, and the assignment error) can be attributed to noisy pseudo labeling. Another question arises: *how to combat noise inherent with pseudo labels and facilitate semi-supervised learning?* To answer this, we estimate the noise by quantifying the uncertainty of noise-polluted region proposals. In particular, we measure the region uncertainty from the perspective of incorrect or imprecise predictions, then involve this uncertainty for noise-resistant semi-supervised learning.

In this paper, we propose a region uncertainty quantification based semi-supervised learning method for object detection. Specifically, we first present a detailed investigation into the effects of noisy pseudo labels. We observe that different types of region proposals behave with different sensitivity in the face of noisy labels. By associating the varied sensitivity with noisy pseudo labels, we present a quantitative metric to measure the uncertainty degrees of regions and construct an uncertainty-aware soft target as the learning objective. In addition, we remove the competitive effect among classes to allow multi-peak probabilistic confidence and avoid over-confident predictions for uncertain regions. By quantifying and embracing the region uncertainty, we obtain a novel semi-supervised learning approach for object detection with high noise resistance and superior performance.

Our main contributions can be summarized as follows:

- We investigate the negative effects of noisy pseudo labels. A novel region uncertainty quantification metric is proposed from the perspective of the sensitivity to noisy pseudo labeling.
- We propose a noise-resistant semi-supervised learning approach by formulating an uncertainty-aware soft target as the learning objective, which prevents the performance from deterioration caused by noisy pseudo labeling.
- By removing the competition among classes and allowing multi-peak probability distributions, we further alleviate the overfitting to noisy pseudo labels.

Our method achieves the state-of-the-art results on the PASCAL VOC and MS COCO dataset, exceeding supervised baseline methods by 6.2% and 4.2% respectively.

## 2   Related Work

**Semi-supervised learning.**  To reduce the difficulty of obtaining large-scale labeled data, semi-supervised learning is presented by adopting unlabeled images into training. Until recently, current

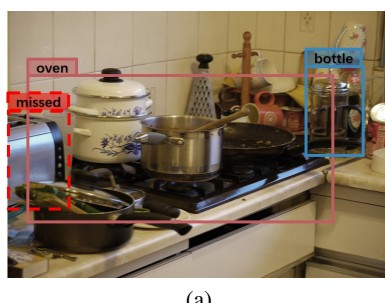 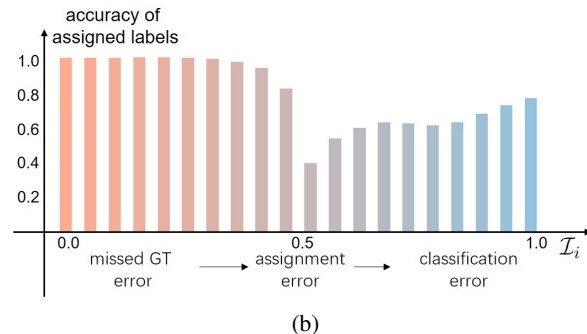

| (a) | (b) |

Figure 2: **Illustration of different types of noise within pseudo labels (a) and their effects on different types of region proposals (b).** Noisy pseudo labels mainly cause the missed GT error, the classification error and the assignment error to hurt the model's performance. These effects behave differently on regions according to their different IoUs with pseudo labels.

semi-supervised methods are mainly about consistency regularization [16, 26, 40, 28], which usually performs data augmentation or perturbation first then constrains their output to be consistent; self-training [44, 4, 30], which usually adopts the co-training mechanism of teacher and student models; label propagation [48, 2] that helps improve the quality of pseudo labels and so on. These methods usually target at classification or semantic segmentation. Object detection, however, is by nature different. Large numbers of region proposals within a single image and the existence of the background category make these methods hard to transfer well to object detection.

**Semi-supervised object detection.** Object detection, one of the most important tasks in computer vision, has developed rapidly in recent years [9, 8, 33, 32, 23]. But the large annotation cost restricts the scale of current detection datasets, further limits the performance of detection models. Currently, a part of semi-supervised object detection methods are based on pseudo labels [31, 24, 43, 39, 47]. But they usually fail to consider noise within pseudo labels thus are easy to overfit noisy labels. Some methods also adopt the idea of data augmentation and consistency regularization [12, 13, 37]. But they usually need to adopt extensive forms of data augmentation for consistency regularization and increase the training budget. We mainly focus on the usage of pseudo labels. Compared with these methods, we target a noise-robust semi-supervised learning approach when using pseudo labels.

**Uncertainty-based pseudo-labeling.** Considering that standard deep learning methods do not possess the ability to model its uncertainty, a series of methods [7, 25, 17] have been proposed. They usually aim at estimating how uncertain a model is about its predictions. Since uncertainty estimation is strongly related to the quality of pseudo labels, some methods [27, 34, 46] have adopted it for pseudo label based semi-supervised learning. Recent works [14, 42] also try to improve the calibration of the network in object detection. However, these works usually focus on measuring the uncertainty of the pseudo label itself. Object detection, in comparison, is different, as pseudo labels do not supervise the learning process directly. Instead, assigned labels for different regions are the learning targets. Therefore, we focus on uncertainty quantification for regions in object detection.

## 3 Region Uncertainty Quantification

In this section, we present the quantification metric for estimating the uncertainty of region proposals. Specifically, we investigate different forms of noise in pseudo labels and observe that different types of proposals possess different noise resistance. Base on this fact, we quantify the uncertainty of regions based on their sensitivity to noisy pseudo labeling.

### 3.1 Noisy Labeling Effects in Semi-supervised Object Detection

Most existing methods for semi-supervised object detection are through pseudo labels. For an unlabeled image, its pseudo labels are usually extracted by a detection model pre-trained on fully-labeled images. To improve the quality of pseudo labels, we perform data distillation [31] on

unlabeled images. Specifically, different data transformations are applied to the unlabeled images for test augmentation. Detection results from images under different transformations are then averaged to form the final pseudo labels $G = \{bb_n, c_n, s_n\}_{n=1}^N$, where $bb_n$ denotes the bounding box, $c_n$ is its category label, and $s_n$ is the corresponding probability score from the classification branch.

For an unlabeled image, after the region proposal network (RPN) [33], a series of region proposals are generated. A certain number of region proposals are then randomly sampled, which forms a set of region proposals $\{r_i\}$ for the later classification and regression. For a region proposal $r_i$, its label is assigned according to its overlaps with pseudo labels. We consider the maximum value among it: $\mathcal{I}_i = \max_{g \in G} \mathrm{IoU}(r_i, g)$. If $\mathcal{I}_i$ is larger than a pre-defined lower bound threshold $\Delta_b$, it is regarded as a positive proposal, and the corresponding label $\{bb_i, c_i, s_i\}$ from the pseudo label set $G$ will be assigned. Otherwise, it is regarded as a negative one. Its category $c_i$ will be assigned as background, while the bounding box $bb_i$ and the probability score $s_i$ are absent.

In the semi-supervised object detection setting, noise inherently exists in pseudo labels. The assigned category $c_i$ is thus likely to be incorrect. The origin of incorrect assigned labels is mainly three-fold:

1. *Missed GT error*. The proposal $r_i$ contains an object, but the corresponding annotation is missed in pseudo labels $G$, like the red dashed `toaster` bounding box in Fig. 2a. $r_i$ fails to match with any annotations thus is assigned with the background category.
2. *Classification error*. The categories of pseudo labels may be incorrect, like the blue 'bottle' bounding box in Fig. 2a, which should be a `cup`. Consequently, although the proposal $r_i$ is able to be assigned with the correct object, its category $c_i$ is incorrect.
3. *Assignment error*. Some pseudo labels are inaccurately localized, like the oversized `oven` in Fig. 2a, so $\mathcal{I}_i$ is erroneous. For the proposal $r_i$, inaccurate bounding box labels in $G$ make it easily be assigned to another object or the background category, resulting in the incorrectly assigned $c_i$.

These three types of errors have different effects on different types of region proposals. According to their definitions, the classification error only happens on positive proposals, while the missed GT error only occurs on negative ones. The missed GT error only matters when the specific negative region is close enough to a miss-annotated object. However, since the number of negative regions is quite large, the noisy negative bounding box is a little improbable to be randomly sampled. As a result, in the region proposal set $\{r_i\}$, it is much less likely for a negative proposal to be affected by the missed GT error, compared to positive proposals under the classification error. We thus conclude that *negative proposals suffer from less noise than positive ones*.

The assignment error affects region proposals whose $\mathcal{I}$ is close to $\Delta_b$ most severely. For these proposals, even a slight localization error of pseudo labels may change the value of $\mathcal{I}$, then disturb the comparison of $\mathcal{I}$ and $\Delta_b$. The assignment error thus occurs and leads to the wrong category label. Therefore, we hold that *proposals whose $\mathcal{I}$ is close to $\Delta_b$ are exposed to more noise*.

We conduct a baseline semi-supervised object detection experiment on the COCO dataset [22], extract all region proposals on unlabeled images, and record whether their assigned labels are correct. The relation between the accuracy of their assigned labels and $\mathcal{I}$ is plotted in Fig. 2b. We observe that the accuracy of negative proposals is universally larger than that of positive ones, and proposals whose $\mathcal{I}$ are close to $\Delta_b$ (0.5) are more noisy. This result confirms our analysis above.

### 3.2 Uncertainty Quantification

According to the above section, as the assigned category $c_i$ for the proposal $r_i$ may be incorrect, the region proposals are kind of noisy. However, since the image is unlabeled, we do not know whether the proposal is noisy or not. This causes region proposals to be uncertain. Current models lack strategies to estimate the uncertainty degrees of regions, not to mention avoiding noisy ones. In this section, we present a quantification metric for the uncertainty of region proposals.

The uncertainty of a region proposal $r_i$ is closely related to whether its assigned category $c_i$ is correct. If the category $c_i$ is likely to be incorrect, the proposal $r_i$ is also uncertain for the model. The definition of uncertainty quantification, therefore, should be consistent with the likelihood that the corresponding region proposal is noisy. According to section 3.1, negative proposals are less likely to be noisy than positive ones. Also, since background samples are significantly more, it is universally acknowledged that foreground samples occupy a more important position in training

[21, 3]. Therefore, we only quantify the uncertainty for positive proposals, and assume that negative proposals are all accurate and certain. Positive proposals are likely to be noisy because of the classification error and the assignment error. We thus need to take both errors into consideration for the uncertainty quantification.

For the proposal $r_i$ and its assigned pseudo label $\{bb_i, c_i, s_i\}$, the classification error derives from the incorrect $c_i$. $s_i$, the classification probability score, can well reflect its self-confidence, thus can be used for measuring the classification error. Through test time augmentation, a simple but effective strategy for assessing the uncertainty of the classification results, $s_i$ contains more information thus is more accurate for evaluating the classification error.

The assignment error is closely related to the overlaps between the proposal and all pseudo labels - the distance between $\mathcal{I}_i$ and the threshold $\Delta_b$. So we adopt $\mathcal{I}_i$ to measure this. Since our uncertainty quantification only aims at positive samples, only region proposals whose $\mathcal{I}_i$ is larger than $\Delta_b$ is considered. Plus, the assignment error is prominent for proposals with a close-to-$\Delta_b$ $\mathcal{I}_i$. We thus set an upper bound threshold $\Delta_f$ and only consider proposals with $\mathcal{I}_i$ in the range from $\Delta_b$ to $\Delta_f$. In this range, we normalize the $\mathcal{I}_i$ to $0 \sim 1$ with a *sigmoid* mapping function, with $\mathcal{I}_i^n$ denoting the normalized $\mathcal{I}_i$. We think $\mathcal{I}_i^n$ is a reasonable metric for estimating the assignment error:

$$\mathcal{I}_i^n = \begin{cases} 1/(1 + \exp(-C \cdot \mathcal{I}_i^r)) & \text{if} \quad \Delta_b < \mathcal{I}_i < \Delta_f \\ 1 & \text{otherwise} \end{cases} \tag{1}$$

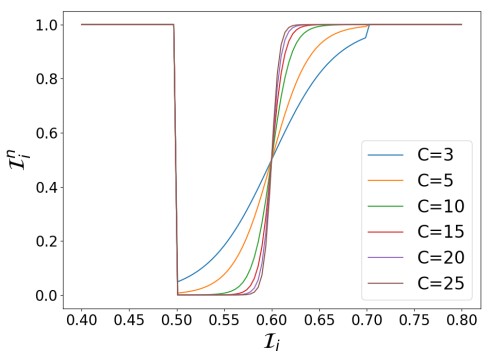

Figure 3: $\mathcal{I}_i^n$, the normalized maximum overlap, w.r.t. $\mathcal{I}_i$, the maimum overlap, under different C. $\Delta_b$ is 0.5 and $\Delta_f$ is 0.7.

where $\mathcal{I}_i^r$ is the linear normalization of $\mathcal{I}_i$ from $\Delta_b \sim \Delta_f$ to $-1 \sim 1$. This ensures the symmetry of $\mathcal{I}_i^n$. $C$ is a pre-defined hyper-parameter. $\mathcal{I}_i^n$ for proposals whose $\mathcal{I}_i$ is not in the range from $\Delta_b$ to $\Delta_f$ is set to 1 manually, since we assume that the assignment error has little effect on them. From Fig. 3, we can see that only if $C$ is not too small, $\mathcal{I}_i^n$ is approximately continuous for positive proposals. For proposals whose $\mathcal{I}_i$ is quite close to $\Delta_b$, its value is quite small, since these proposals are much more likely to be affected by the assignment error. Then, $\mathcal{I}_i^n$ increases as $\mathcal{I}_i$ increases, until $\mathcal{I}_i$ is close to $\Delta_f$. At this time, $\mathcal{I}_i^n$ is close to 1 and the assignment error can be neglected. As we can see, $\mathcal{I}_i^n$ is in the range $0 \sim 1$ and is negatively correlated to the assignment error, thus can be used for estimating the assignment error.

Based on the above analysis, $s_i$ and $\mathcal{I}_i^n$ can be utilized for assessing the classification error and the assignment error separately. A region is certain only when both the classification error and assignment error are small. We thus multiply them to consider both errors. Then, we propose our uncertainty quantification $u_i$ for the proposal $r_i$, as follows:

$$u_i = \begin{cases} 1 - s_i \cdot \mathcal{I}_i^n & \text{if} \quad \mathcal{I}_i > \Delta_b \\ 0 & \text{otherwise} \end{cases} \tag{2}$$

For a positive proposal, its uncertainty metric $u_i$ is relevant to its $s_i$ and $\mathcal{I}_i^n$. If both of them are close to 1, the possibility that it is prone to the classification error and the assignment error is small. This means that its assigned category $c_i$ is relatively accurate and it is somewhat certain. So its $u_i$ is close to 0, according to Equ. 2, vice versa. For negative proposals, as we presume that they are less likely to be noisy, they are certain to the model. So their uncertainty quantification is set to 0 manually.

## 4   Noise-Resistant Semi-supervised Learning

When region proposals are uncertain, current detection models lack means of handling uncertainty. They cannot discriminate uncertain regions or alleviate uncertainty during training, which makes it easy to overfit noise. In this section, we propose to introduce uncertainty into semi-supervised learning, so that it better adapts to uncertain regions. Finally we obtain noise-resistant learning.

## 4.1 Constructing Uncertainty-aware Soft Target

In object detection, for the proposal $r_i$, after assigned with the label $\{bb_i, c_i, s_i\}$, classification and regression are conducted on this single proposal. Generally, *softmax* function is used to convert the output logits into the probability distribution $\boldsymbol{p}_i = [p_{ik}]$, where $p_{ik}$ denotes the probability prediction of the proposal $r_i$ for the $k$-th category. *Cross entropy* loss is then utilized for classification: $L_{cls} = CE(\boldsymbol{y}_i, \boldsymbol{p}_i)$. $\boldsymbol{y}_i = [y_{ik}]$ is the one-hot category label for $r_i$, where $y_{ic_i} = 1$ and others are 0. When $\boldsymbol{p}_i = \boldsymbol{y}_i$, the cross entropy is minimized. At this time, the probability distribution $\boldsymbol{p}_i$ is also one-hot and its certainty is maximized.

In fully-supervised object detection, $c_i$ is always the actual ground truth category of $r_i$, denoted with $c_i^{gt}$, so this is of no problem. But in the semi-supervised setting, assigned labels may be incorrect and $c_i \neq c_i^{gt}$ may occur. When $c_i \neq c_i^{gt}$, the network still tries to maximize $p_{ic_i}$ so that $\boldsymbol{p}_i$ can approach $\boldsymbol{y}_i$. The only outcome is that the cross entropy between $\boldsymbol{p}_i$ and the actual one-hot label of $r_i$ is enlarged, which increases the actual loss. As a result, the optimized $\boldsymbol{p}_i$ approximates $\boldsymbol{y}_i$, the distribution of pseudo labels. The model thus overfits to noise, which degrades its performance.

The above phenomenon occurs because when region proposals are uncertain, the model still tries to optimize itself towards a certain objective. This disparity causes the model hard to accommodate the uncertain regions. Since the assigned label $\{bb_i, c_i, s_i\}$ for $r_i$ is uncertain, just adopting the hard label $\boldsymbol{y}_i$ is unreasonable. To address this issue, we utilize our proposed uncertainty metric $u_i$, and construct a soft target label for the proposal $r_i$, denoted with $\hat{\boldsymbol{y}}_i = [\hat{y}_{ik}]$:

$$\hat{y}_{ik} = \begin{cases} (1 - u_i(\beta))y_{ik} & \text{if} \quad k \neq \text{bg (background)} \\ 1 - \sum_{k \neq \text{bg}} \hat{y}_{ik} & \text{otherwise} \end{cases} \tag{3}$$

where

$$u_i(\beta) = \begin{cases} 1 - (s_i \cdot \mathcal{I}_i^n)^\beta & \text{if} \quad \mathcal{I}_i > \Delta_b \\ 0 & \text{otherwise} \end{cases} \qquad \beta = (t/T)^q \tag{4}$$

For foreground categories, the soft target is calculated by multiplying its certainty quantification $1 - u_i$ and hard label $y_{ik}$. The soft target of the background category is then obtained by restricting the summation of $\hat{\boldsymbol{y}}_i$ to 1. For a positive proposal $r_i$, if it is relatively certain and its assigned label $c_i$ is trustworthy, its $u_i$ is close to 0, and $\hat{\boldsymbol{y}}_i$ is close to $\boldsymbol{y}_i$. Since it is certain, the hard label $\boldsymbol{y}_i$ is of no problem. If the label $c_i$ for $r_i$ is very likely to be incorrect, its uncertainty quantification $u_i$ is high, close to 1. Then the soft label $\hat{\boldsymbol{y}}_i$ approaches the hard label of negative proposals. In this way, the network regards $r_i$ as a background category. Since negative samples are noise-resilient, they are less likely to be interfered with by these uncertain samples. With this action, positive proposals with higher quality are involved in classification. Noisy and uncertain regions do not participate in the training of positive samples, thus mitigate the noise overfitting problem.

$\beta$ in Equ. 3 and Equ. 4 is a dynamic exponent, where $t$ denotes the current number of iteration and $T$ is the total number of iterations. $q$ is a pre-defined parameter to make $\beta$ increase in a polynomial manner. The introduction of $\beta$ is to make the soft target $\hat{\boldsymbol{y}}_i$ dynamically change with the semi-supervised training process. For deep learning networks under noisy labels, they will usually learn from clean and easy samples at the beginning, and overfit to noisy labels later [1, 45, 10]. In the beginning, $\beta$ is close to 0 thus $u_i(\beta)$ is close to 0 too. At this time, $\hat{\boldsymbol{y}}_i$ is basically the same as $\boldsymbol{y}_i$ and the training is just like previous methods, which guarantees enough amount of positive data for the initial training. Later, $\beta$ starts to increase thus $u_i(\beta)$ also increases. $\hat{\boldsymbol{y}}_i$ begins to change and provides high-quality samples when the model is gradually easy to overfit to noise. With this schedule, the soft target $\hat{\boldsymbol{y}}_i$ can better adapt to the training.

Finally, the soft target label participates in the learning. For the classification branch, we utilize the KL divergence. For the regression branch, we choose the commonly used L1 loss, and re-weight it with the dynamic uncertainty quantification metric $1 - u_i(\beta)$:

$$L_{cls} = \text{KL}(\hat{\boldsymbol{y}}_i || \boldsymbol{p}_i), \qquad L_{reg} = (1 - u_i(\beta))L_1 \tag{5}$$

## 4.2 Promoting Cross-category Uncertainty

The above section aims to solve the problem by avoiding the hard label $\boldsymbol{y}_i$. However, besides the hard label $\boldsymbol{y}_i$, the probability distribution $\boldsymbol{p}_i$ is also problematic. $\boldsymbol{p}_i$ is usually obtained with the

Table 1: **Experimental Results on PASCAL VOC 2007 test for different methods.** FS is the supervised model. We re-implement DD and CSD method based on the Faster RCNN.

| Method | Labeled | Unlabeled | $AP_{50:95}$ | $AP_{50}$ | $AP_{75}$ |
|---|---|---|---|---|---|
| FS | VOC07 | - | 43.1 | 76.9 | 43.1 |
| DD [31] | VOC07 | VOC12 | 45.2 | 77.8 | 47.1 |
| CSD [12] | VOC07 | VOC12 | 45.0 | 77.5 | 46.7 |
| STAC [37] | VOC07 | VOC12 | 44.6 | 77.5 | - |
| ours | VOC07 | VOC12 | **49.3** | **80.6** | **53.0** |
| FS | VOC0712 | - | 52.9 | 83.9 | 57.6 |
| DD | VOC07 | VOC12 + coco-20cls | 46.6 | 79.0 | 49.0 |
| CSD | VOC07 | VOC12 + coco-20cls | 45.5 | 77.4 | 47.8 |
| STAC | VOC07 | VOC12 + coco-20cls | 46.0 | 79.1 | - |
| ours | VOC07 | VOC12 + coco-20cls | **50.2** | **81.4** | **54.2** |

*softmax* function, which restricts the summation of $\boldsymbol{p}_i$ to 1. This generates a single-peak distribution $\boldsymbol{p}_i$, where $p_{ic_i}$ reaches its highest value. The entropy of such probability distribution is usually small, as the strong lateral inhibition effect of *softmax* represses the cross-category uncertainty of $\boldsymbol{p}_i$. When region proposals are uncertain, this kind of probability distribution is also inappropriate. If $c_i \neq c_i^{gt}$, enlarging $p_{ic_i}$ means a smaller $p_{ic_i^{gt}}$, finally resulting in overfitting to incorrect categories.

To avoid the lateral inhibition effect of *softmax*, we adopt *sigmoid* for outputting the probability distribution $\boldsymbol{p}_i$. With *sigmoid* function, the cross-category uncertainty is thus promoted and the multi-peak distribution $\boldsymbol{p}_i$ is allowed. In this way, raising $p_{ic_i}$ does not necessarily mean suppressing $p_{ic_i^{gt}}$ when $c_i \neq c_i^{gt}$. The probability distribution $\boldsymbol{p}_i$ turns more uncertain since its entropy is enlarged under *sigmoid*, which better adapts to the uncertain region proposals. We finally choose to use focal loss [21] to promote the cross-category uncertainty, which adopts *sigmoid* for probability output.

Based on focal loss, we further introduce our constructed soft target $\hat{\boldsymbol{y}}_i$ into semi-supervised learning. Following [41], we propose to use the soft-target focal loss with $\hat{\boldsymbol{y}}_i$ as the soft target. In this way, uncertainty is introduced into both the target and the probability distribution, making the learning more noise-resistant.

## 5 Experiment

### 5.1 Experimental Setting

**Dataset.** We mainly conduct our proposed method on PASCAL VOC [6] and MS COCO [22]. For simple notation, we refer to a subset containing 35k images from COCO 2014 validation set as coco-35, the training set with 80k images as coco-80, and their union set as coco-115. The 120k images from COCO 2017 unlabeled set are denoted with coco-120. We also select images from COCO trainval consisting of only 20 categories as PASCAL VOC, and denote these 19,592 images as coco-20cls. We mainly adopt four settings: 1) VOC07 trainval (5,011 images) as labeled set and VOC12 trainval (11,540 images) as unlabeled set; 2) VOC07 trainval as labeled set, VOC12 trainval and coco-20cls as unlabeled set; 3) coco-35 as labeled set and coco-80 as labeled set; 4) coco-115 as labeled set and coco-120 as unlabeled set. Besides, we also follow previous methods for experiments with a gradually increasing percentage of labeled examples.

**Implementation Details.** We conduct our experiment using Pytorch [29] and MMDetection [5]. Unless otherwise specified, we use Faster RCNN [33] with ResNet50 [11] and FPN [20]. The standard 1x schedule is adopted. The generation of pseudo labels and their filtration are basically the same as that in [31]. For hyper-parameters, we set $\Delta_b$ to 0.5. $\Delta_f$ is set to 0.7 first, and changes to 0.8 after the first decay of learning rate. $C$ is set to 15 and $q$ is set to 0.1.

### 5.2 Comparison with Existing Methods

**PASCAL VOC.** We perform the comparative study with Faster RCNN on the VOC dataset. The results are presented in Tab. 1. The fully-supervised detection model trained on VOC07 trainval obtains a 43.1% $AP$ and 76.9% $AP_{50}$. When using the raw pseudo labels without consideration

Table 2: **Experimental Results on COCO minival for different methods.** FS is the supervised model. We re-implement DD method based on the Faster RCNN. † denotes adopting strong data augmentation in training.

| Method | Labeled | Unlabeled | $AP_{50:95}$ | $AP_{50}$ | $AP_{75}$ |
|---|---|---|---|---|---|
| FS | coco-35 | - | 31.3 | 52.0 | 33.0 |
| DD [31] | coco-35 | coco-80 | 33.1 | 53.3 | 35.4 |
| ours | coco-35 | coco-80 | **35.5** | **54.5** | **38.7** |
| FS | coco-115 | - | 37.4 | 58.1 | 40.4 |
| DD | coco-115 | coco-120 | 37.9 | 60.1 | 40.8 |
| PL [38] | coco-115 | coco-120 | 38.4 | 59.7 | 41.7 |
| CSD [12] | coco-115 | coco-120 | 38.9 | - | - |
| STAC † [37] | coco-115 | coco-120 | 39.2 | - | - |
| Ubteacher † [24] | coco-115 | coco-120 | 41.3 | - | - |
| Humble teacher † [39] | coco-115 | coco-120 | 42.4 | - | - |
| Instant-teaching † [47] | coco-115 | coco-120 | 40.2 | - | - |
| ours (1x) | coco-115 | coco-120 | 40.6 | 59.7 | 44.4 |
| ours (3x) | coco-115 | coco-120 | **41.7** | **61.0** | **45.9** |
| ours † (3x) | coco-115 | coco-120 | **43.2** | **62.0** | **47.5** |

Table 3: **Experimental Results on COCO minival with a gradually increasing percentage of labeled examples.** All methods except CSD adopt strong data augmentation in training.

| Method | 1% COCO | 2% COCO | 5% COCO | 10% COCO |
|---|---|---|---|---|
| CSD [12] | $10.51 \pm 0.06$ | $13.93 \pm 0.12$ | $18.63 \pm 0.07$ | $22.46 \pm 0.08$ |
| STAC [37] | $13.97 \pm 0.35$ | $18.25 \pm 0.25$ | $24.38 \pm 0.12$ | $28.64 \pm 0.21$ |
| Ubteacher [24] | $17.84 \pm 0.12$ | $21.98 \pm 0.07$ | $26.30 \pm 0.22$ | $29.64 \pm 0.10$ |
| Humble teacher [39] | $16.96 \pm 0.38$ | $21.72 \pm 0.24$ | $27.70 \pm 0.15$ | $31.61 \pm 0.28$ |
| Instant-teaching [47] | $18.05 \pm 0.15$ | $22.45 \pm 0.15$ | $26.75 \pm 0.05$ | $30.40 \pm 0.05$ |
| ours | $\mathbf{18.41 \pm 0.10}$ | $\mathbf{24.00 \pm 0.15}$ | $\mathbf{28.96 \pm 0.29}$ | $\mathbf{32.43 \pm 0.20}$ |

of noise, the method (i.e. DD) obtains a 45.2% $AP$ and 77.8% $AP_{50}$. Our method, in comparison, achieves a 49.3% $AP$ and 80.6% $AP_{50}$. The final $AP$ outperforms the baseline method by more than 4% and also performs significantly better than other methods. This large margin strongly demonstrates the effectiveness of our method. Compared to other methods, performance metrics from $AP_{50}$ to $AP_{75}$ show consistent improvement, which proves that our method better adapts to uncertain region proposals in semi-supervised learning and achieves a more noise-resistant training.

Further, we augment the unlabeled dataset by adding images from coco-20cls. With more data available, the performance is further improved to 50.2% $AP$ and 81.4% $AP_{50}$. This illustrates the usefulness of our proposed method on datasets of different distribution, which is usually an inevitable problem when utilizing large-scale datasets. Also, this proves the potential of semi-supervised object detection to break through the upper bound of fully-supervised object detection.

**MS COCO.** We also evaluate our method on the COCO dataset, a more challenging dataset, and the results are listed in Tab. 2. In the coco-35/80 setting, compared to the baseline method DD, our method achieves a 2.4% AP improvement, which is prominent for the COCO dataset. In the coco-115/120 setting, where a larger-scale dataset is adopted, we observe that previous methods obtain higher accuracy compared to that on PASCAL VOC, which indicates that more available data is beneficial to semi-supervised learning. And we notice that our approach also achieves better results. Even with a 1x schedule, our method still outperforms many current methods, which usually adopt a 3x schedule and more training time. We further adopt 3x schedule and the same data augmentation strategy as that in [24, 39, 47]. Our method achieves a better result, a 43.2% AP, 0.8% more than the state-of-the-art, which further indicates the superiority of our method.

To further demonstrate the effectiveness of our method, we follow the same setting as that in previous methods [37, 24, 39, 47] with different degrees of supervision using the COCO dataset. For a fair comparison, we adopt the same data augmentation strategy. The results are listed in Tab. 3. For unbiased teacher [24] which uses larger batch size and longer training schedules, we retrain it under

Table 4: **Ablation Study on VOC07 test.** CCU: cross-category uncertainty, UST: uncertainty-aware soft target.

| Setting | CCU | UST | $AP_{50:95}$ | $AP_{50}$ | $AP_{75}$ |
|---|---|---|---|---|---|
| fully | | | 43.1 | 76.9 | 43.1 |
| | ✓ | | 42.5 | 77.2 | 41.6 |
| semi | | | 45.2 | 77.8 | 47.1 |
| | ✓ | | 46.2 | 79.4 | 47.9 |
| | | ✓ | 48.2 | 78.3 | 52.2 |
| | ✓ | ✓ | 49.3 | 80.6 | 53.0 |

Table 5: **Ablation Study on COCO.** CCU: cross-category uncertainty, UST: uncertainty-aware soft target.

| Setting | CCU | UST | $AP_{50:95}$ | $AP_{50}$ | $AP_{75}$ |
|---|---|---|---|---|---|
| fully | | | 31.3 | 52.0 | 33.0 |
| | ✓ | | 31.2 | 52.5 | 32.9 |
| semi | | | 33.1 | 53.3 | 35.4 |
| | ✓ | | 34.3 | 55.3 | 36.4 |
| | | ✓ | 33.7 | 51.1 | 37.0 |
| | ✓ | ✓ | 35.5 | 54.5 | 38.7 |

the common training schedules with the released official implementation. It is noteworthy that our method outperforms CSD [12] and STAC [37] by a large margin. Compared to recent works such as unbiased teacher [24], instant-teaching [47] and humble teacher [39], our uncertainty-aware noise-resistant learning can also consistently achieve better results under different degrees of supervised data. Particularly under the 2% supervision data setting, our method outperforms the instant-teaching by 1.6%. With 5% supervision data, our method outperforms humble teacher by 1.2%. This comparative study further validates the performance of our proposed method.

## 5.3 Ablation Study

We perform ablation study on VOC and COCO to analyze the impact of our designation. The results are listed in Tab. 4 and Tab. 5, where we use VOC07/12 and COCO35/80 setting separately. We do not adopt any data augmentation strategy in this section.

**Cross-category Uncertainty.** From Tab. 4, we notice that the enhancement of promoting cross-category uncertainty is limited in the fully-supervised setting. $AP_{50:95}$ and $AP_{75}$ even decrease a little. This is reasonable. In the fully-supervised setting, region proposals are certain, so suppressing the cross-category uncertainty is no problem. In addition, in two-stage detectors, the extreme class-imbalance problem is mollified by RPN and random sampling, leaving little space for focal loss to improve by hard example mining. But in the semi-supervised setting, it is effective. By promoting cross-category uncertainty, the lateral inhibition effect is mitigated and the probability of the actual category is increased, which better accommodates the uncertain regions. It brings about a 1.6% $AP_{50}$ improvement. The more precise classification result even helps the localization, increasing $AP_{50:95}$ by 1.0% and $AP_{75}$ by 0.8%. From Tab. 5, a similar trend is also observed. This again demonstrates that promoting cross-category uncertainty is effective for semi-supervised detection.

**Uncertainty-aware Soft target.** From Tab. 4, we observe that after adopting the uncertainty-aware soft target and optimizing with the KL divergence, the overall $AP$ is improved by 3.0%. The localization ability is boosted even more - $AP_{75}$ increases by over 5%. This is because under the soft target, those certain regions contribute more to the classification for positive proposals and bounding box regression. With the uncertainty quantification involved in the soft target, the network better avoids uncertain regions. After further combining with the focal loss, the performance is further boosted, finally achieving a 49.3% $AP$ and 80.6% $AP_{50}$. This illustrates that adopting uncertainty into both the probability distribution and classification target can further accommodate the uncertain region proposals. Experimental results from the COCO dataset in Tab. 5 also verify its effectiveness.

**Visualization of Noise Quantification.** We perform illustrative analysis to explain the dynamic uncertainty quantification $u(\beta)$. Specifically, we collect positive proposals that are assigned with the correct labels (clean ones) and the wrong labels (noisy ones). The probability distribution of their $u(\beta)$ is plotted in Fig. 4, where Fig. 4a, 4b, 4c is the distribution in the early, middle, late stage separately. We observe that clean region proposals generally have lower $u(\beta)$ than noisy ones, which demonstrates that our designed $u(\beta)$ is reasonable for uncertainty quantification. From Fig. 4a, we notice that most clean proposals and about a half of noisy proposals possess small $u(\beta)$. This ensures enough data for the model's initial training. As the training proceeds, the value of $u(\beta)$ turns larger because of the dynamic $\beta$, from the middle stage of Fig. 4b to the late stage of Fig. 4c. At this time, almost all noisy samples possess a close-to-1 $u(\beta)$. As a cost, $u(\beta)$ for many clean proposals is also enlarged. But some of them are still preserved. This guarantees that the model is learned with high-quality proposals when it is sensitive to noise.

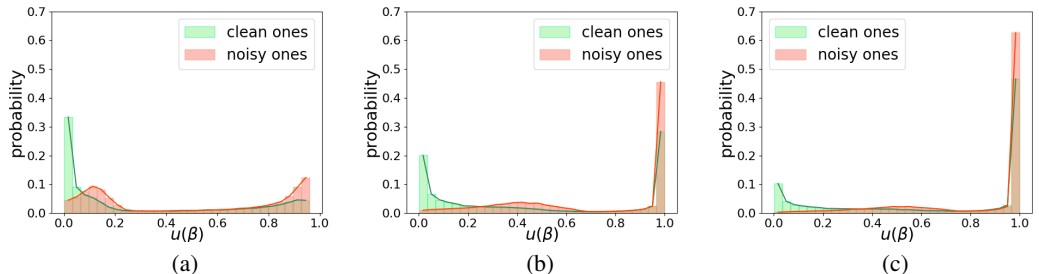

Figure 4: **Distribution of the uncertainty quantification** $u(\beta)$ **on clean or noisy region proposals** in the early stage of training (a), the middle stage (b) and the late stage (c). Clean regions' $u(\beta)$ is usually smaller than noisy ones, and $u(\beta)$ increases gradually as the training proceeds.

**The choice of hyper-parameters.** We also evaluate our method under different hyper-parameter settings, including the value of $C$, $\Delta_f$, $q$. The results are in the supplementary material. We find that our method is relatively robust to the variance of hyper-parameters. Therefore, even in a new environment, it is not so hard to tune the hyper-parameters.

## 6 Conclusion

In this paper, we focus on pseudo label based semi-supervised learning and target at combating noise. By utilizing our proposed uncertainty quantification as the soft target and facilitating multi-peak probability distribution, we introduce uncertainty into semi-supervised learning. Experimental results on multiple semi-supervised object detection settings demonstrate that our proposed method can better adapt to the uncertain regions, thus achieves the state-of-the-art results. In the future, we will further explore more efficient approaches to utilize unlabeled data.

## Acknowledgments and Disclosure of Funding

This work was supported by the National Natural Science Foundation of China under Grant No. 61771288, Cross-Media Intelligent Technology Project of Beijing National Research Center for Information Science and Technology (BNRist) under Grant No. BNR2019TD01022 and the research fund under Grant No. 2019GQG0001 from the Institute for Guo Qiang, Tsinghua University.

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
