# Supplementary Material for "Combating Noise: Semi-supervised Learning by Region Uncertainty Quantification"

## 1 Qualitative Results

We perform qualitative results in this section to further illustrate our method. We use Data Distillation [3] as our baseline method.

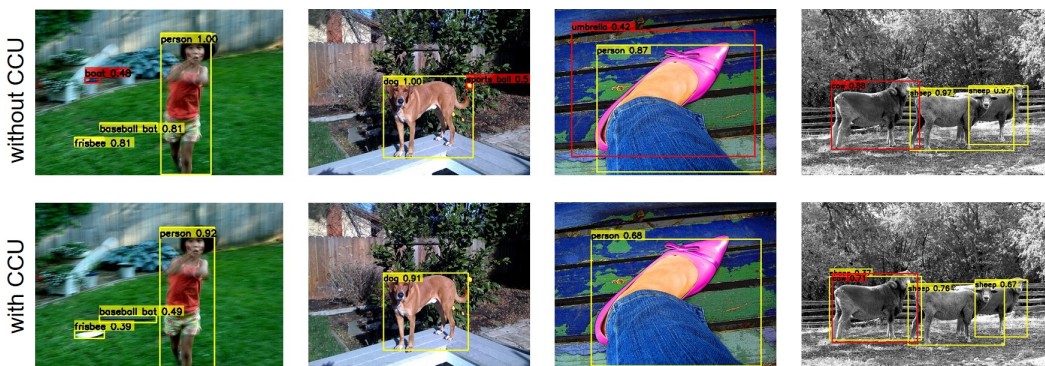

Figure 1: **Illustrative examples for images with or without cross-category uncertainty.** The first row is the results of the baseline method, and the second row is from the method after promoting cross-category uncertainty. Red bounding boxes are incorrect predictions, and yellow bounding boxes are correct predictions. The incorrect predictions are repressed, while the correct predictions are encouraged.

**Cross-category Uncertainty.** To further explain the effectiveness of promoting cross-category uncertainty, we present illustrative examples in Fig. 1. As we analyze in our paper, the main effect of promoting cross-category uncertainty is to alleviate over-confident category predictions under noisy labeling. The illustrative examples well validate this point. In the first couple of examples, we observe that the semi-supervised baseline model detects a part of the background as a boat. After cross-category uncertainty is promoted, the probability of the incorrect category 'boat' is suppressed. The same thing also happens in the second and the third example, where the baseline detector incorrectly detects a sports ball or an umbrella. These problems are also addressed by promoting cross-category uncertainty. In the fourth example, the baseline model misclassifies the left sheep as a cow, maybe because its visual appearance is confusing in this image. After cross-category uncertainty is promoted, we notice that the probability of the incorrect category 'cow' is lowered, while the score of the correct category 'sheep' is increased. As a result, a correct instance bounding box appears. These examples demonstrate that **promoting cross-category uncertainty helps repress the probability of incorrect categories and boost the probability of correct categories** in the semi-supervised setting.

**Uncertainty-aware soft target.** We also present examples in Fig. 2 to illustrate the function of the uncertainty-aware soft target. In the first example, besides the correct predictions, the baseline model

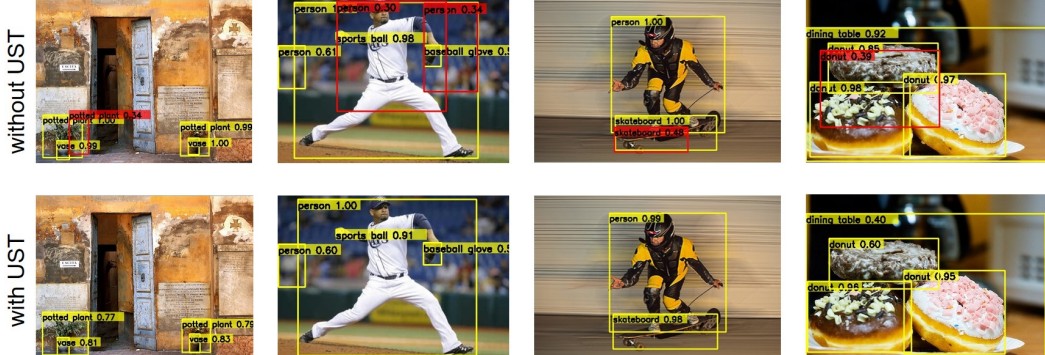

Figure 2: **Illustrative examples for images with or without uncertainty-aware soft target.** The first row is the results of the baseline method, and the second row is from the method after adopting uncertainty-aware soft target. Red bounding boxes are incorrect predictions, and yellow bounding boxes are correct predictions. The redundant predictions are removed because of the better classification and regression ability.

detects a small part of the potted plant as a plant, which is so small that it should be a background region. This error may derive from two reasons. The first is that the classification branch of the model is not perfect, so the detector classifies a background region as a potted plant. The second is that the model classifier the region correctly, but regresses it to the wrong position. NMS cannot remove this kind of duplicate bounding box, resulting in an incorrect prediction. After uncertainty-aware soft target is adopted, the detector benefits from more certain regions during training. Both the classification ability and the regression ability are boosted, a result of combating noise. As a result, we can find that the redundant potted plant is removed. This also applies to the other examples. In the second example, a local part of the person is detected and a background region is also detected as a person incorrectly. In the third example, a part of the skateboard is detected. And in the fourth example, an unexpected donut appears. After using the uncertainty-aware soft target, these errors are all corrected. This demonstrates that **uncertainty-aware soft target contributes to the better classification and regression performance.**

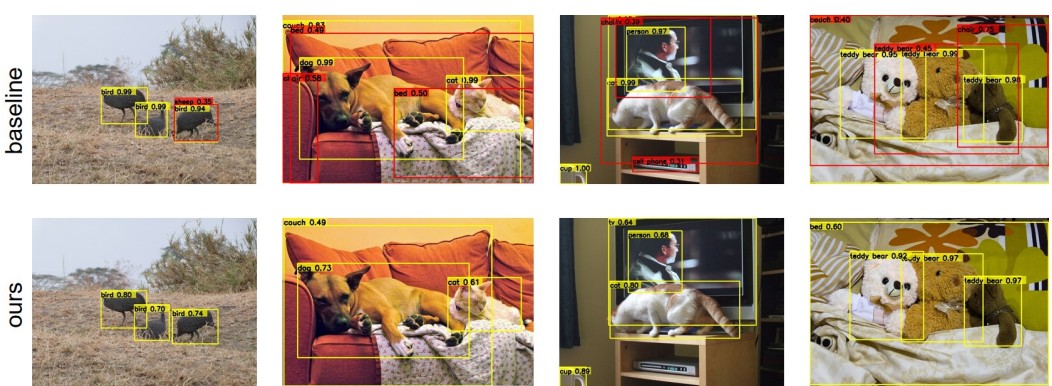

Figure 3: **Comparative examples on different images.** The first row is from the baseline method, and the second row is from our method. Red bounding boxes are incorrect predictions, and yellow bounding boxes are correct predictions. With uncertainty involved in semi-supervised learning, our method achieves better results.

**Overall results.** Finally, we present images in Fig. 3 to compare our method with the baseline method. In the first example, the baseline detector is confused about the right bird, so that a sheep bounding box is also detected. In the second example, many incorrect results such as 'bed' or 'chair' are detected. The third example is also the same, where mistakes include the 'chair' and the 'cell phone'. In the fourth example, redundant teddy bear, incorrect chair and couch are detected. The overfitting to incorrect categories, the classification error or the localization error give rise to these mistakes. Our method involves uncertainty into semi-supervised learning, thus **alleviates overfitting**

**to incorrect categories and generates better classification and regression results**. As a result, our method manages to solve the above detection mistakes and produce better results.

## 2 Quantitative Results

<table>
<tr><td colspan="4">Table 1: Varying $q$ on VOC07 test.</td></tr>
<tr><td>$q$</td><td>$AP_{50:95}$</td><td>$AP_{50}$</td><td>$AP_{75}$</td></tr>
<tr><td>0 (static)</td><td>48.1</td><td>79.9</td><td>51.4</td></tr>
<tr><td>0.01</td><td>48.4</td><td>80.0</td><td>51.4</td></tr>
<tr><td>0.1</td><td>49.3</td><td>80.6</td><td>53.0</td></tr>
<tr><td>0.5</td><td>48.8</td><td>80.1</td><td>52.3</td></tr>
<tr><td>1.0</td><td>48.2</td><td>79.9</td><td>51.4</td></tr>
</table>

<table>
<tr><td colspan="4">Table 2: Varying $q$ on COCO minival.</td></tr>
<tr><td>$q$</td><td>$AP_{50:95}$</td><td>$AP_{50}$</td><td>$AP_{75}$</td></tr>
<tr><td>0 (static)</td><td>35.0</td><td>53.7</td><td>38.4</td></tr>
<tr><td>0.01</td><td>35.3</td><td>54.4</td><td>38.4</td></tr>
<tr><td>0.1</td><td>35.5</td><td>54.5</td><td>38.7</td></tr>
<tr><td>0.5</td><td>35.5</td><td>54.5</td><td>38.7</td></tr>
<tr><td>1.0</td><td>35.2</td><td>54.1</td><td>38.5</td></tr>
</table>

**The choice of $q$ in Equ. 4.** According to Equ. 4 in our paper, the value of $q$ controls the increase pattern of $\beta$ in a polynomial manner: $\beta = (t/T)^q$. $\beta$ acts as a dynamic exponent for the uncertainty quantification metric $u(\beta)$ to make it better adapt to the semi-supervised learning.

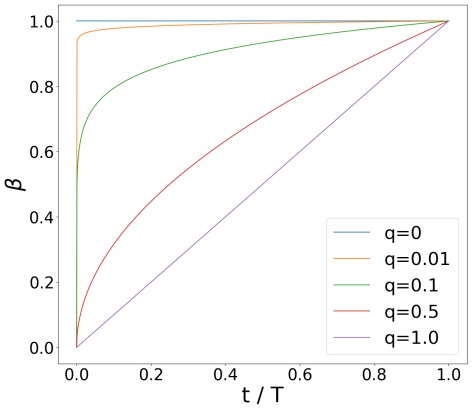

Figure 4: $\beta$ w.r.t. $t/T$ under different $q$.     Figure 5: $\mathcal{I}_i^n$ w.r.t. $\mathcal{I}_i$ under different C.

We perform a series of experiments for the choice of $q$ on the PASCAL VOC [1] and MS COCO [2] dataset. The results are listed in Tab. 1 and Tab. 2. The variance of $\beta$ with $t/T$ under different $q$ is plotted in Fig. 4. When $q$ equals 0, the value of $\beta$ is always 1. At this time, the uncertainty quantification metric $u$ is static. From Tab. 1, we notice that the static $u$ generates a 48.1% $AP$ and 79.9% $AP_{50}$. In comparison, without the uncertainty-aware soft target, the model achieves 46.2% $AP$ and 79.4% $AP_{50}$, which can be seen in the ablation study section of our paper. This proves that **our uncertainty-aware soft target helps the network better avoid uncertain regions**.

Despite this, the static $q$ ignores that deep neural networks possess different noise sensitivity in the different training stages. In the early stage, the network learns easy knowledge and is relatively noise-resilient. At this time, $\beta$ can be small, so that enough number of positive regions participate in training. In the late stage, the network has fitted to most samples, so $\beta$ should be large to avoid uncertain regions. As a result, to adapt to the semi-supervised learning, $\beta$ should increase as the training proceeding. As a polynomial exponent, $q$ should be larger than 0. Also, most of the increase patterns, such as exponential increase or logarithmic increase, can be approximated in a polynomial manner. Therefore, the polynomial increase $\beta$ can accommodate most situations.

From Tab. 1 and Tab. 2, we notice that when $q$ is 0.1, the results are the best. $\beta$ increases quite rapidly at the beginning, then it turns slower later. This indicates that for better adapting to the learning, $\beta$ should increase quickly at the beginning. Its value is quite small when the training starts, to guarantee enough positive regions. Then, it increases rapidly, to quickly avoid uncertain regions, until $\beta$ turns large and the increased speed turns small. If $q$ is smaller, $\beta$ increases too quickly in the beginning,

making positive regions scarce. If $q$ is larger, it increases slow, and uncertain regions cannot be avoided sufficiently.

Also, from Tab. 1 and Tab. 2, we find that **the results are relatively robust to the choice of $q$**. Especially for the COCO dataset, the $AP$ divergence from a 0.01 $q$ to a 0.5 $q$ is less than 0.2%. This indicates that our designed uncertainty quantification metric $u$ is reasonable. Therefore, the uncertainty-aware soft target is able to help detection models avoid uncertain regions even under different $q$.

<table>
<tr><td colspan="4" align="center">Table 3: Varying $C$ on VOC07 test.</td></tr>
<tr><td>$C$</td><td>$AP_{50:95}$</td><td>$AP_{50}$</td><td>$AP_{75}$</td></tr>
<tr><td>3</td><td>48.6</td><td>80.2</td><td>52.1</td></tr>
<tr><td>10</td><td>49.0</td><td>80.5</td><td>52.1</td></tr>
<tr><td>**15**</td><td>**49.3**</td><td>**80.6**</td><td>**53.0**</td></tr>
<tr><td>20</td><td>49.0</td><td>80.4</td><td>52.8</td></tr>
<tr><td>25</td><td>49.0</td><td>80.2</td><td>52.8</td></tr>
</table>

<table>
<tr><td colspan="4" align="center">Table 4: Varying $C$ on COCO minival.</td></tr>
<tr><td>$C$</td><td>$AP_{50:95}$</td><td>$AP_{50}$</td><td>$AP_{75}$</td></tr>
<tr><td>3</td><td>35.2</td><td>54.4</td><td>38.2</td></tr>
<tr><td>10</td><td>35.5</td><td>54.5</td><td>38.4</td></tr>
<tr><td>**15**</td><td>**35.5**</td><td>**54.5**</td><td>**38.7**</td></tr>
<tr><td>20</td><td>35.4</td><td>54.5</td><td>38.6</td></tr>
<tr><td>25</td><td>35.4</td><td>54.4</td><td>38.6</td></tr>
</table>

**The choice of $C$ in Equ. 1.** To calculate the quantification metric $\mathcal{I}_i^n$ for the assignment error, a sigmoid function with a hyper-parameter $C$ is used, according to Equ. 1. in our paper. The variance of $\mathcal{I}_i^n$ under different $C$ is plotted in Fig. 5. From Tab. 3 and Tab. 4, we notice that when $C$ is quite small, such as 3, the obtained $AP$ is low. This is up to our expectation, since at this time, the figure of $\mathcal{I}_i^n$ is even not continuous. The discontinuous point is obvious at $\Delta_b$ and $\Delta_f$. The break point is not reasonable, which affects the performance. When $C$ is a little larger, the figure of $\mathcal{I}_i^n$ is approximately continuous. Then, the performance gets better. We thus choose $C = 15$ in our experiment. When $C$ gets larger, $\mathcal{I}_i^n$ is almost binary. At this time, $\mathcal{I}_i^n$ is almost a hard quantification for the assignment error by thresholding $\mathcal{I}_i$. So the performance is reduced. Despite this, we still observe that **our results are relatively robust to the choice of $C$**.

**The choice of $\Delta_b$.** $\Delta_b$ denotes the threshold that dividing positive region proposals and negative ones. We adopt the usual setting for it, where $\Delta_b$ equals to 0.5.

<table>
<tr><td colspan="4" align="center">Table 5: Varying $\Delta_f$ on VOC07 test.</td></tr>
<tr><td>$\Delta_f$</td><td>$AP_{50:95}$</td><td>$AP_{50}$</td><td>$AP_{75}$</td></tr>
<tr><td>0.6</td><td>48.3</td><td>80.5</td><td>51.5</td></tr>
<tr><td>0.7</td><td>48.6</td><td>80.5</td><td>51.7</td></tr>
<tr><td>0.8</td><td>48.6</td><td>79.8</td><td>51.6</td></tr>
<tr><td>0.9</td><td>48.0</td><td>78.7</td><td>51.6</td></tr>
</table>

<table>
<tr><td colspan="4" align="center">Table 6: Varying $\Delta_f$ on COCO minival.</td></tr>
<tr><td>$\Delta_f$</td><td>$AP_{50:95}$</td><td>$AP_{50}$</td><td>$AP_{75}$</td></tr>
<tr><td>0.6</td><td>34.9</td><td>54.5</td><td>37.6</td></tr>
<tr><td>0.7</td><td>35.3</td><td>54.8</td><td>38.6</td></tr>
<tr><td>0.8</td><td>35.1</td><td>54.0</td><td>38.3</td></tr>
<tr><td>0.9</td><td>34.5</td><td>52.7</td><td>37.5</td></tr>
</table>

**The choice of $\Delta_f$.** $\Delta_f$ is the upper bound for estimating the assignment error. For region proposals whose $\mathcal{I}_i$ is larger than $\Delta_f$, we think that they are a little unlikely to be affected by the assignment error. Therefore, $\Delta_f$ should be a value between $\Delta_b$ and 1. We perform experiments under different $\Delta_f$, the results are listed in Tab. 5 and Tab. 6. We observe that when $\Delta_f$ is 0.7, the performance is the best. If $\Delta_f$ is smaller, such as 0.6, the quantification for the assignment error is quite insufficient. This is because only proposals whose $\mathcal{I}_i$ is in the range from 0.5 to 0.6 are processed. This range is quite small. As a result, only a small fraction of regions is estimated. Most of uncertain regions are regarded as certain ones. They participate in the training and affect the semi-supervised learning, which hurts the performance. If $\Delta_f$ is larger, such as 0.9, the model believes that the assignment error may affect almost all positive regions. The model thus lacks enough positive samples for training.

Since a 0.7 $\Delta_f$ generates the best performance, we choose this value for training. Considering deep neural networks possess different noise sensitivity in the different training stage, we make $\Delta_f$ dynamic too. In the early stage, it is small to guarantee enough samples. In the late stage, it is large to provide high-quality proposals for training. In object detection, learning rate is usually decayed after 70% and 90% of the total number of iterations. When the first decay happens, the performance usually increases significantly. In the semi-supervised setting, this also means that the model begins to overfit to noise significantly. Therefore, we increase the value of $\Delta_f$ after the first learning rate decay. The results on the change of $\Delta_f$ are listed in Tab. 7 and Tab. 8. We notice that if $\Delta_f$ increases

Table 7: The value change of $\Delta_f$ on VOC07.

| $\Delta_f$ | $AP_{50:95}$ | $AP_{50}$ | $AP_{75}$ |
|---|---|---|---|
| $0.7 \rightarrow 0.7$ | 48.6 | 80.5 | 51.7 |
| $0.7 \rightarrow 0.75$ | 49.2 | 80.6 | 52.4 |
| **$0.7 \rightarrow 0.8$** | **49.3** | **80.6** | **53.0** |
| $0.7 \rightarrow 0.9$ | 49.3 | 80.0 | 52.9 |

Table 8: The value change of $\Delta_f$ on COCO.

| $\Delta_f$ | $AP_{50:95}$ | $AP_{50}$ | $AP_{75}$ |
|---|---|---|---|
| $0.7 \rightarrow 0.7$ | 35.3 | 54.8 | 38.6 |
| $0.7 \rightarrow 0.75$ | 35.4 | 54.7 | 38.7 |
| **$0.7 \rightarrow 0.8$** | **35.5** | 54.5 | **38.7** |
| $0.7 \rightarrow 0.9$ | 35.3 | 53.8 | 38.8 |

after the first decay, the performance is indeed better. Therefore, in our experiment, we set $\Delta_f$ to 0.7 first, then increase it to 0.8.