# OpenReview forum: "Combating Noise: Semi-supervised Learning by Region Uncertainty Quantification"
_NeurIPS.cc/2021/Conference — NeurIPS 2021 Poster_

### Official Review · Reviewer_123Z · 2021-07-12

**Rating:** 6
**Confidence:** 5

**Summary:**

This submission addresses the semi-supervised object detection task. Compared to previous methods, the authors propose to quantify the uncertainty degree and apply it into the model training. Besides, the authors modify the classification error to allow multi-peak in probability distribution from soft label targets. The experiments on four settings of semi-supervised object detection show the model achieves good performances.

**Limitations And Societal Impact:**

Please check the "weakness" part mentioned above

**Main Review:**

### Strengths
* The motivation and technical novelties are clearly explained. The design of the uncertainty from noisy labels is reasonable and can be easily plugged into other two-stage semi-supervised object detectors' training

* The authors provide full details of hyper-parameter ablation study in both submission and supplementary materials, which is much appreciated.

### Weakness
* The novelty is incremental in engineering. The design of noisy labels' uncertainty can be only applied into the scenario of two-stage object detectors' training. For other one-stage detectors or anchor-free detectors, there may be no $\Delta_b$ calculated from IOU, thus limited the scope of the novelty.

* The experiment results are not significant to show the effectiveness of the design. The compared the method is not the state-of-the-art performance. Before the NeuRIPS submission deadline, there is an ICLR 2021 paper [R1]. The reported performance on Table 1 VOC+coco-20cls is 50.34 $AP_{50:95}$, which is higher than the proposed model's performance. In Table 2, the reported performance on coco-115 + coco-120 is 41.3 $AP_{50:95}$, with only 0.4 decrease compared to that from the proposed model.

[R1]: Unbiased Teacher For Semi-supervised Object Detection, ICLR 2021.

**Time Spent Reviewing:**

3

---

> ### Author Response · Authors · 2021-08-10
> **We have addressed the concern about the novelty in engineering and the experiment results.**
>
> We thank the reviewer for the positive feedback and constructive feedback. We provide the response in the following paragraphs.
>
> **1. "The novelty is incremental in engineering. The design of noisy labels' uncertainty can be only applied into the scenario of two-stage object detectors' training."**
>
> * Our method can be easily extended to one-stage detectors or anchor-free detectors. For one-stage detectors, anchors can be regarded as regions, and the calculation of IoU-based uncertainty is basically the same as that in two-stage detectors. For anchor-free detectors, such as CenterNet, the position of bounding box will be regressed with the key points. The regressed bounding box can also be used for the calculation of IoU and uncertainty quantification. We conduct a preliminary experiment on the VOC dataset with the one-stage RetinaNet, and achieves the $AP$ of 46.1\%, 0.9\% higher than the baseline method (45.2\% of $AP$). This preliminary experiment demonstrates the feasibility and novelty scope of our method.
>
> * Since most of existing semi-supervised object detection methods, including DD, STAC, unbiased teacher conduct the experiments based on two-stage detectors, partially for superior performance, we follow this common experimental setting for better and fair comparison.
>
> **2. "The experiment results are not significant to show the effectiveness of the design."**
>
> We would like to state that some recent works such as unbiased teacher [R1] adopt strong data augmentation strategy during training. The main results in our submitted paper are obtained without any data augmentation, which is better than current methods without data augmentation. When the same augmentation strategy is adopted, our method can obtain 51.5\% $AP$ on VOC+coco-20cls dataset and 43.2\% $AP$ on the coco-115/120 dataset. Compared to unbiased teacher (newly published in ICLR2021), the $AP$ is increased by 1.2\% on the VOC+coco-20cls dataset and nearly 2\% on the coco-115/120 dataset. In particular, the improvement on the large-scale coco-115/120 setting is considered as significant. More comprehensive comparison with existing methods in the table below further demonstrates the effectiveness of the proposed method.
>
> |   Method  |  Dataset | data augmentation  | $AP_{50:95}$ |
> | :---------: | :--------: |:--------: | :------: |
> |DD | VOC + coco-20cls |  | 46.6 |
> |CSD | VOC + coco-20cls |  | 45.5 |
> |ours | VOC + coco-20cls |  | 50.2 |
> |STAC | VOC + coco-20cls |  $\checkmark$ | 46.0 |
> |unbiased teacher [R1] | VOC + coco-20cls | $\checkmark$ | 50.3 |
> |ours | VOC + coco-20cls | $\checkmark$ | **51.5** |
> |DD | coco-115 + coco-120 |  | 37.9 |
> |PL | coco-115 + coco-120 |  | 38.4 |
> |CSD | coco-115 + coco-120 |  | 38.9 |
> |ours | coco-115 + coco-120 |  | 41.7 |
> |STAC | coco-115 + coco-120 |  $\checkmark$ | 39.2 |
> |unbiased teacher [R1] | coco-115 + coco-120 |  $\checkmark$ | 41.3 |
> |ours | coco-115 + coco-120 |   $\checkmark$ | **43.2**  |
>
> [R1] "Unbiased Teacher for Semi-Supervised Object Detection", ICLR 2021

---

### Official Review · Reviewer_jvYC · 2021-07-15

**Rating:** 6
**Confidence:** 3

**Summary:**

This paper tackles semi-supervised learning for object detection, where only a small amount of the training data is labeled with bounding boxes, while the rest of the training data is unlabeled. To solve this problem, the authors present a region uncertainty quantification method that quantifies the uncertainty of regions in unlabeled images based on their sensitivity to noisy pseudo labels. In particular, a quantification metric for the uncertainty is proposed for positive region proposals according to the classification error and the assignment error. Experiments are performed on both PASCAL VOC and MS COCO, with a comparison to three competitors: DD, CSD, STAC.

**Limitations And Societal Impact:**

Discussion about limitations and possible negative societal impacts are not provided in the paper. It is suggested to add an individual section at the end of the paper.

**Main Review:**

The paper an interesting and practically important problem in semi-supervised object detection. The materials in this paper are overall well-orgnanized and well-presented. Weakness regarding the methodology and experiments are summarized in the following.

**Method**

Although the paper addresses an interesting and practically important problem in semi-supervised object detection, the writing could be improved. In particular, the method section is hard to follow, several concerns are listed below.

1. The self-confidence score is estimated by averaging probability scores from different data-augmented images. However, such estimation is not directly related to data distillation in [25]. Distillation is not related to how the confidence score derives in the paper. It is therefore suggested not to mention data distillation in line 155, but instead explain why one should averaging multiple augmented images to derive the confidence score.
2. The definition of Eq. (1) is unclear. Why does it quantify the assignment errors in region proposals?
3. The motivation of Eq. (2) is also unclear. Why should the self-confidence scores multiply the uncertainty of assignment I?
4. It is unclear how the focal loss in [18] helps to promote cross-category uncertainty. It is suggested to write down the equation and analyze how the soft targets are used in the focal loss.

**Experiments**
The experiments are well designed to show the effectiveness on different datasets, under different use of the unlabeled data. In the following, several suggestions are given.
1. The evaluation of the competitors (DD, CSD, STAC) is unclear. It is suggested to add a paragraph to describe these competitors in Section 5.1. In particular, a discussion about the difference of the proposed method as compared to the competitors is suggested.
2. In the ablation study, it is suggested to analyze the necessity of bringing the different uncertainty estimators (1) uncertainty on classification vs. (2) uncertainty on assignment. It is unclear how these estimators contribute to the model.



**Time Spent Reviewing:**

3

---

> ### Author Response · Authors · 2021-08-10
> **We have addressed the concern about the writing and experiment**
>
> We appreciate the reviewer’s time and effort for providing constructive feedback and insightful questions.
>
> **1. "It is therefore suggested not to mention data distillation in line 155, but instead explain why one should averaging multiple augmented images to derive the confidence score."**
>
> Thank you for the suggestion. We will replace the mentioning of data distillation with the explanation of averaging multiple augmented images in the revised version. We average multiple augmented images to derive the confidence score because the self-confidence score can be biased. In some cases, the prediction might be wrong but the confidence score is still high. Averaging over test time augmentation is a simple yet effective strategy for assessing the uncertainty of the networks' classification results and improving the precision of self-confidence score.
>
> **2. "The definition of Eq. (1) is unclear. Why does it quantify the assignment errors in region proposals?"**
>
> * The assignment error derives from inaccurately localized pseudo labels. Because when the pseudo labels are inaccurately localized, the IoU based label assignment for different region proposals can be disturbed. The assignment error is closely related to the maximum IoU between the object and pseudo labels. We normalize the IoU to 0 $\sim$ 1 with a non-linear function, and that is Eq. (1). Since it's positively related to IoU, its relative value can well reflect the assignment error. Also, the figure of the non-linear function (Fig. 3) displays a similar trend with the accuracy in Fig. 1, which indicates that Eq. (1) is reasonable to measure the assignment error.
>
> * For example, considering a region that is quite close to an object in pseudo labels, it is relatively impossible to be assigned to an incorrect object. This means its assignment error is small. Since it is close to a pseudo label, its maximum IoU with the pseudo label is large. After being normalized with Eq. (1), the outcome will be close to 1, a high score. Therefore, the result of Eq. (1) can quantify the assignment error.
>
> **3. "The motivation of Eq. (2) is also unclear. Why should the self-confidence scores multiply the uncertainty of assignment I?"**
>
> A region is certain only when both the classification error and assignment error are small. The certainty metric is high only when the self-confidence score is high ($s_i$ is close to upper limit 1) and the assessment of assignment error is small ($\mathcal{I}_i^n$ is close to upper limit 1). Motivating by this, we use the multiplication of $s_i$ and $\mathcal{I}_i^n$ to quantify the certainty. Therefore the uncertainty quantification is further formulated as $1-s_i \mathcal{I}_i^n$.
>
> **4. "It is unclear how the focal loss helps to promote cross-category uncertainty. It is suggested to write down the equation and analyze how the soft targets are used in the focal loss."**
>
> * Focal loss adopts *sigmoid* for converting the scores into probabilities. Compared to *softmax*, *sigmoid* alleviates the lateral inhibition effect and allows multi-peak probability distribution. Compared to the single-peak probability caused by *softmax* activation, the cross-category uncertainty is promoted by allowing multi-peak distribution. This can better adapt to the semi-supervised learning.
>
> * To integrate the soft target into training, we use the soft target focal loss. Its equation is as follows:
> \begin{equation}
>     L_{sfl} = -  \alpha_t(\hat{y}) \cdot |\hat{y} - p|^\gamma \cdot ( \hat{y} {\rm log}(p_t) + (1-\hat{y}) {\rm log}(1-p))
> \end{equation}
> where
> \begin{equation}
> \alpha_t(\hat{y}) =
> \alpha \hat{y} +  (1 - \alpha)(1-\hat{y}) \\ {\rm if} \\ y_k>0
> \end{equation}
> With this kind of soft target focal loss, the network will be optimized towards the soft target $\hat{y}$, which meets our expectation and fulfill the same purpose as the KL-divergence.
>
> **5. "The evaluation of the competitors (DD, CSD, STAC) is unclear. It is suggested to add a paragraph to describe these competitors in Section 5.1. In particular, a discussion about the difference of the proposed method as compared to the competitors is suggested."**
>
> Thank you for the suggestion. We will add the discussion in the final version. Our method is significantly different from existing works. In general, DD directly uses pseudo labels for semi-supervised object detection. CSD adopts consistency regularization which is widely used in classification for semi-supervised detection. STAC focuses on the data augmentation in semi-supervised training. By studying different types of noise in pseudo labels and measuring them with region uncertainty quantification, we propose a noise-robust semi-supervised learning approach. Experimental results show that our method outperforms DD by 4.1\% on the VOC dataset, which indicates the effectiveness of our proposed uncertainty quantification and noise-robust training. Also, our method is 4.3\% higher than CSD and 4.7\% than STAC, which further demonstrates its effectiveness.
>
> **6. "In the ablation study, it is suggested to analyze the necessity of bringing the different uncertainty estimators (1) uncertainty on classification vs. (2) uncertainty on assignment. It is unclear how these estimators contribute to the model."**
>
> To analyze the effectiveness of each uncertainty estimator, we perform the ablation study on the VOC dataset. The detection accuracy is listed in the below table. Without uncertainty estimation in soft target, the detector obtains 46.2\% $AP$ and 79.4\% $AP_{50}$. When the uncertainty on assignment is introduced, $AP$ is 48.3\% and $AP_{50}$ is 79.5\%. Moreover, when the uncertainty on classification is introduced, we achieve $AP$ is 49.3\% and $AP_{50}$ is 80.6\%. The results demonstrate the effectiveness of the two different uncertainty estimators. From this table we can see that the uncertainty on assignment contributes more to the localization ability ($AP_{50:95}$ from 46.2\% to 48.3\%), while the uncertainty on classification contributes more to the classification performance ($AP_{50}$ from 79.5\% to 80.6\%).
>
> |   uncertainty on assignment  |  uncertainty on classification  | $AP_{50:95}$ | $AP_{50}$ |
> | :---------: | :--------: | :------: | :------: |
> | | |46.2 | 79.4|
> | $\checkmark$ | |48.3 | 79.5|
> | $\checkmark$| $\checkmark$|49.3 | 80.6|
>
> **7. Limitations And Societal Impact**
>
> We will further discuss the limitation and societal impact in the final version. As semi-supervised learning requires large-scale unlabeled data, the training budget may increase. We also recognize that the technology might lead to privacy concerns if deployed for surveillance.

---

### Official Review · Reviewer_1uTd · 2021-07-16

**Rating:** 6
**Confidence:** 5

**Summary:**

**Goal:**
This paper aims to address semi-supervised object detection, where the object is trained with a set of labeled images (fully-annotated box and object category) and a set of unlabeled images.

**Method:**
- The authors propose to use both classification score and IoU to pseudo-box to compute the uncertainty of each proposal boxes, and this is based on their analysis that assignment errors are large when the IoU is closer to 0.5.

- Rather than using the hard label for the classification branch, they construct the soft labels by using uncertainty-based to weigh the ground-truth label.

- The author also proposes to use focal loss and sigmoid to alleviate the overfitting issue of incorrect categories.

**Experiments:**
- They present the results on VOC and COCO datasets and show the improvement against CSD and STAC.


**Limitations And Societal Impact:**

The authors did not mention the potential negative societal impact in the paper.

**Main Review:**

**Pros:**
- Analyses and categorization of errors in pseudo-labels are clear and reasonable, and, based on the observation, the authors further proposed to use IoU to construct the region uncertainty.

**Cons:**
- The authors did not mention and compare to the prior works, including Unbiased Teacher [1], Instant-Teaching [2], or Humble Teacher [3], as they show the significant improvement against CSD and STAC.

- In the prior works of semi-supervised object detection, one important experimental benchmark is to examine the effectiveness of methods under different degrees of supervision using COCO-dataset (i.e., 1%-10% of COCO images as labeled and the remaining images as unlabeled set). This important experiment is missing in the paper, and this makes the proposed method hard to be compared with the existing works on semi-supervised object detection tasks.

- As the proposed method has more hyper-parameters (such as C in Eq.1, beta and q in Eq. 4) and some hyper-parameter scheduling than the prior works, this makes the proposed method hard to be directly applied to new datasets or environments. These hyper-parameters might need to be tuned for the new datasets.


[1] "Unbiased Teacher for Semi-Supervised Object Detection", ICLR 2021

[2] "Instant-Teaching: An End-to-End Semi-Supervised Object Detection Framework", CVPR 2021

[3] "Humble Teachers Teach Better Students for Semi-Supervised Object Detection", CVPR 2021

**Time Spent Reviewing:**

6 hours

---

> ### Author Response · Authors · 2021-08-10
> **We have compared results with recent works, conducted experiments under different degrees of supervision using COCO-dataset and addressed the concern about hyper-parameters.**
>
> We thank the reviewer for the constructive feedback about result comparison, experiment and hyper-parameters. We address the concern below.
>
> **1. "The authors did not mention and compare to the prior works, including Unbiased Teacher, Instant-Teaching, or Humble Teacher, as they show the significant improvement against CSD and STAC."**
>
> * Thank you for the suggestion. We miss the comparison with the mentioned three works mainly because they are published near the NeurIPS submission. We will add the related discussion and comparative experiments in the final version.
>
> * We conduct the comparative experiments on the coco-115/120 dataset. For fair comparison, we adopt the same data augmentation schedule. The detection accuracy is listed in the table below. It can be seen from the table below that our method still shows superior performance over the recent published works. Our method outperforms unbiased teacher [1] by 1.9\%. Also, we are 3\% higher than instant-teaching [2] and 0.9\% higher than humble teacher [3]. This superior performance mainly derives from our proposed uncertainty-aware noise-resistant training, which can effectively combat the noise in pseudo-label based semi-supervised learning. Our work is also a different practice from current methods for semi-supervised object detection.
>
> |    Method   |  $AP_{50:95}$ |
> | :---------: | :--------: |
> |unbiased teacher (ICLR 2021) [1] | 41.3 |
> |instant-teaching (CVPR 2021) [2] | 40.2 |
> |humble teacher (CVPR 2021) [3] | 42.3 |
> |ours | **43.2** |
>
> **2. "one important experimental benchmark is to examine the effectiveness of methods under different degrees of supervision using COCO-dataset (i.e., 1\%-10\% of COCO images as labeled and the remaining images as unlabeled set)."**
>
> We conduct the comparative experiments under different degrees of supervision on COCO-dataset. The detection accuracy is listed in the table below. For unbiased teacher [1] which uses larger batch size and longer training schedules, we retrain it under the common training schedules with the released official implementation. It is noteworthy that our method outperforms CSD and STAC by a large margin. Compared to recent works such as unbiased teacher [1], instant-teaching[2] and humble teacher [3], our uncertainty-aware noise-resistant learning can also consistently achieve better results under different degrees of supervised data. Particularly under the 2\% supervision data setting, our method outperforms the instant-teaching by 1.6\%. With 5\% and 10\% supervision data, our method outperforms humble teacher by 1.2\%. This comparative study further validates the performance of our proposed method and we will add it in the final version.
>
> |    Method   |  1\% COCO  | 2\% COCO | 5\% COCO | 10\% COCO |
> | :---------: | :--------: | :------: | :------: | :-------: |
> |     CSD (NeurIPS 2019)    |  10.2  |   13.6   | 18.9     |    24.5   |
> |     STAC (Arxiv 2020)    |  13.9  |  18.2   | 24.3    |   28.6   |
> |     unbiased teacher (ICLR 2021} [1] |  17.8  |   21.9   | 26.3     |   29.6   |
> |     instant-teaching (CVPR 2021) [2] |  18.0  |   22.4   | 26.7    |   30.4   |
> |     humble teacher (CVPR 2021) [3] |  16.9  |   21.7   | 27.7    |   31.6   |
> |     ours     |  **18.4**  |   **24.0**   |  **28.9**     |    **32.8**   |
>
> **3. "The proposed method has more hyper-parameters and some hyper-parameter scheduling than the prior works. This makes the proposed method hard to be directly applied to new datasets or environments."**
>
> We perform ablation study on hyper-parameters and additional experiments, which demonstrates that hyper-parameters are not an obstacle to new datasets or environments.
>
> * Due to page limit, we present the impact of hyperparameters over different datasets in the submitted supplementary material (as in Section 2, Table 1-8 of supplementary material). According to the experiments, the detection performance of our proposed method is relatively robust to the choice of hyper-parameters. In particular, the performance almost remains the same on the COCO dataset when $C$ in Eq.(1) varies from 3 to 25 and when $q$ varies from 0.01 to 1.0 ($\beta$ is automatically adjusted by $q$). Therefore, it is not so hard to tune the hyper-parameters.
>
> * The current setting of hyper-parameters can also be directly applied to new datasets. For example, experimental results in our submitted paper and this rebuttal material share the same hyper-parameters on VOC and COCO dataset. Also, we perform experiments on the *cityscapes* dataset with the current hyper-parameter setting. The improvement is consistent - we obtain 34.2\% $AP$, 2.1\% higher than the baseline DD method (32.1\% $AP$). The experiments across different datasets show that it is safe to directly apply the reported hyperparameters to new datasets or environments and it is not necessarily tune the hyper-parameters again.
>
> **4. Limitations And Societal Impact.**
>
> We will further discuss the limitation and societal impact in the final version. As semi-supervised learning requires large-scale unlabeled data, the training budget may increase. We also recognize that the technology might lead to privacy concerns if deployed for surveillance.
>
> [1] "Unbiased Teacher for Semi-Supervised Object Detection", ICLR 2021
>
> [2] "Instant-Teaching: An End-to-End Semi-Supervised Object Detection Framework", CVPR 2021
>
> [3] "Humble Teachers Teach Better Students for Semi-Supervised Object Detection", CVPR 2021

---

> > ### Comment · Reviewer_1uTd · 2021-08-22
> > **The rebuttal has addressed my concern**
> >
> > I have read the rebuttal, and it addressed my previous concerns:
> >
> > **[Major issue]** Did not provide an important benchmark (COCO 1%-10%): The authors provided the experiment on COCO-standard, compared to existing works, and also showed the improvement in the rebuttal. Although they only provide one single run for each case, I think it is sufficient to address this issue. I would suggest the author add the COCO-standard experiments with five runs (follow the previous standard benchmark) in the revised paper.
> >
> > [Minor issue] Four hyper-parameters for one single threshold: as the authors showed the hyper-parameters are used for both VOC and COCO and its generalization to Cityscape, the sensitivity of hyper-parameters might not be a concern.
> >
> > I thus raised my rating score to 6 - marginally above the threshold since the authors have addressed my concerns.

---

> > > ### Author Response · Authors · 2021-08-24
> > > **Thank you for your comments**
> > >
> > > Thank you for your positive comments. We will further perform related experiments and add COCO-standard results with five runs in the revised paper.

---

### Official Review · Reviewer_jnH3 · 2021-07-16

**Rating:** 7
**Confidence:** 3

**Summary:**

A method for semi-supervised 2D object detection is proposed. The main contribution is an uncertainty estimate of pseudo-labels that comes from a combination of classification score (s)
and a localization score. The latter is estimated by the IoU score with the set of pseudo-labels and is normalized to values between 0 and 1 only in a certain range. The estimated uncertainty is then utilized in setting the soft target and penalizing the L1 regression loss.

**Limitations And Societal Impact:**

limitation could be further discussed

**Main Review:**

Strengths:
* Experimental results are quite impressive
* The analysis of different failure modes in section 3.1 is pretty interesting.

Weaknesses - major:
* A large effort is devoted to supervising uncertain pseudo-labels. I wonder how important that is. What if all uncertain labels were simply ignored (a-la fix-match)?
* I'm missing a table with a gradually increasing percentage of labeled examples. This is quite common in semi-supervised detection.

minor:
* Please spell out the maximization of IoU with the set G. (max over all g in G, etc)
* Writing could be improved. In particular, section 3 is very wordy and could be more concise. Same goes for e.g lines 247-253 where the idea is repeated using the same terminology instead of explaining from different views or simply explaining once. Also please rephrase sentences like 215 that are vague.
* Confidence of localization was discussed before in [B] and then used in [C] for semi-supervised 3D detection.

missing references:
[A] UNBIASED TEACHER FOR SEMI-SUPERVISED OBJECT DETECTION
[B] Acquisition of Localization Confidence for Accurate Object Detection
[C] 3DIoUMatch: Leveraging IoU Prediction for Semi-Supervised 3D Object Detection

**Time Spent Reviewing:**

2.5

---

> ### Author Response · Authors · 2021-08-10
> **We have clarified the importance of supervising uncertain pseudo labels and provided experimental results with a gradually increasing percentage of labeled examples.**
>
> We thank the reviewer for the positive comments and constructive feedback, and we address the raised issues below.
>
> **1. "A large effort is devoted to supervising uncertain pseudo-labels. I wonder how important that is. What if all uncertain labels were simply ignored (a-la fix-match)?**
>
> To verify the importance of supervising uncertain pseudo labels, we do the comparative study by just ignoring those uncertain labels based on the uncertainty quantification (as in fix-match).
>
> * On the VOC dataset, a 45.9\% $AP$ is achieved with the tuned threshold for ignoring uncertain labels. Compared to baseline (45.2\% of $AP$), less than 0.7\% improvement is obtained. Experiments show that the performance improvement gained by simply ignoring uncertain labels is quite limited. In contrast, with the soft target supervision on uncertain labels, we achieve 48.2\% $AP$, 3\% higher than the baseline and 2.3\% higher than the simply ignoring strategy (as in fix-match). If we adopt focal loss to promote the cross-category uncertainty, the difference is more significant. Simply ignoring uncertain labels only obtains 46.0\% $AP$, which is even a little lower than the baseline method. In comparison, our method obtains 49.3\%, 3.1\% higher than the baseline.
>
> |    Method   |  without CCU | with CCU |
> | :---------: | :--------: | :------: |
> |     baseline     |  45.2   |  46.2 |
> |     ignoring uncertain labels     | 45.9   |  46.0 |
> |     ours     |   **48.2**   | **49.3** |
> Note: CCU denotes cross-category uncertainty in our paper
>
>
> * This is because simply ignoring uncertain labels decreases the number of regions (RoIs) participating in the training. The small amount of RoIs involved in training makes it insufficient to learn a robust detector. In contrast, our proposed method supervises the uncertain regions with soft targets instead of eliminating the RoIs in semi-supervised learning. Therefore, adopting the uncertainty quantification as the soft target for supervising the training is reasonable and important.
>
> **2. "I'm missing a table with a gradually increasing percentage of labeled examples. This is quite common in semi-supervised detection."**
>
> We conduct the experiment with a gradually increasing percentage of labeled examples. The results are listed in the table below. For fair comparison, we adopt the same data augmentation strategy. For unbiased teacher which uses larger batch size and longer training schedules, we retrain it under the common training schedules with the official implementation. It is noticeable that our method achieves consistently superior performance with different percentages of supervised data. Compared to recent work as unbiased teacher [A], our method obtains 2\% higher $AP$ with 2\%, 5\% and 10\% labeled data. Particularly, with 10\% of supervised data, our proposed method outperforms the state-of-the-art by more than 3\%. The experiment with a gradually increasing percentage further demonstrates the effectiveness of our proposed method. We will add the table in the final version.
>
> |    Method   |  1\% COCO  | 2\% COCO | 5\% COCO | 10\% COCO |
> | :---------: | :--------: | :------: | :------: | :-------: |
> |     CSD     |  10.2  |   13.6   | 18.9     |    24.5   |
> |     STAC     |  13.9  |  18.2   | 24.3    |   28.6   |
> |     unbiased teacher [A] |  17.8  |   21.9   | 26.3     |   29.6   |
> |     ours     |  **18.4**  |   **24.0**   |  **28.9**     |    **32.8**   |
>
> **3. "Please spell out the maximization of IoU with the set G."**
>
> Thank you for the suggestion. We will revise it as $ \\max_{g \\in G} {\\rm IoU}(r_i, g)$ in the final version.
>
> **4. "Writing could be improved."**
>
> Thank you for your suggestion. We will carefully improve our writing in the final version, including:
>     1) simplify the writing in Section 3;
>     2) delete the repeated explanations in lines 247-253;
>     3) rewrite line 215, to explain the importance of supervising uncertain regions with soft targets instead of hard labels.
>
> **5. "Confidence of localization was discussed before in [B] and then used in [C] for semi-supervised 3D detection."**
>
> We agree that confidence of localization is discussed in [B] and used in [C]. We will add the related discussion in the final version.
>
> * The motivation of [B] and our work is different. [B] aims to predict localization confidence for refining the detection bounding boxes, while we target at estimating the uncertainty of different regions in semi-supervised learning when pseudo labels are noisy.
>
> * [C] tries to improve the quality of pseudo labels by filtering and ignoring low-quality labels based on the localization confidence for 3D semi-supervised detection. In comparison, we target at combating the noisy pseudo labels with our proposed region uncertainty quantification for noise-resistant semi-supervised learning.
>
> **6. Limitations And Societal Impact**
>
> We will further discuss the limitation and societal impact in the final version. As semi-supervised learning requires large-scale unlabeled data, the training budget may increase. We also recognize that the technology might lead to privacy concerns if deployed for surveillance.
>
> [A] Unbiased Teacher for Semi-Supervised Object Detection, ICLR 2021
>
> [B] Acquisition of Localization Confidence for Accurate Object Detection
>
> [C] 3DIoUMatch: Leveraging IoU Prediction for Semi-Supervised 3D Object Detection

---

### Decision · Program_Chairs · 2021-09-27

**Decision:**

Accept (Poster)

**Comment:**


 As mentioned by multiple reviewers, the ideas in this work and especially analysis of errors/failure modes were found to be interesting and novel. However, the reviewers raised significant legitimate empirical and experimental deficiencies that were surprising. This includes lack of standard experimental settings (e.g. varying amount of labels from 1-10%) and comparisons to recent state of art that were published before the NeurIPS deadline (Unbiased Teacher, Instant-Teaching, and Humble Teacher). Such adherence to standard practices and fair comparisons to related work are necessary to move the field forward and should have been included initially.

  During the rebuttal the authors provided these results and comparisons and addressed the empirical concerns. The authors should add these results, in addition to descriptions of how the proposed method differs from ideas presented in those papers.

  Unfortunately, this did not leave much room for discussion of other aspects of the work. For example, there is a question of novelty (mentioned by 123Z) and furthermore there are several uncertainty-based pseudo-labeling methods (in the context of classification or even segmentation, e.g. [A, B]) that have come out that are not even discussed. The authors should clearly address this and discuss how their methods differ (besides utilizing a similar idea in a new task, object detection).

  Finally, several reviewers pointed out several specific writing improvements that should be addressed. These should be included in the final version.

  Overall, the paper did provide interesting analysis and addressed the experimental concerns, and so can be accepted. However, all of the above must be addressed.

[A] In Defense of Pseudo-Labeling: An Uncertainty-Aware Pseudo-label Selection Framework for Semi-Supervised Learning, ICLR 2021.
[B] Rectifying Pseudo Label Learning via Uncertainty Estimation for Domain Adaptive Semantic Segmentation, IJCV 2021.